# Antibacterial Activity and Sustained Effectiveness of Calcium Silicate-Based Cement as a Root-End Filling Material against *Enterococcus faecalis*

**DOI:** 10.3390/ma16186124

**Published:** 2023-09-08

**Authors:** Seong-Hee Moon, Seong-Jin Shin, Seunghan Oh, Ji-Myung Bae

**Affiliations:** 1Institute of Biomaterials & Implant, College of Dentistry, Wonkwang University, 460 Iksan-daero, Iksan City 54538, Republic of Korea; shmoon06@gmail.com (S.-H.M.); shoh@wku.ac.kr (S.O.); 2Department of Dental Biomaterials, College of Dentistry, Wonkwang University, 460 Iksan-daero, Iksan City 54538, Republic of Korea; ko2742@naver.com

**Keywords:** calcium silicate cement, *Enterococcus faecalis*, antibacterial, sustained antibacterial effectiveness

## Abstract

Several calcium silicate cement (CSC) types with improved handling properties have been developed lately for root-end filling applications. While sealing ability is important, a high biocompatibility and antimicrobial effects are critical. This study aimed to conduct a comparative evaluation of the antimicrobial efficacy and sustained antibacterial effectiveness against *Enterococcus faecalis* (*E. faecalis*) of commercially available CSCs mixed with distilled water (DW) and chlorhexidine (CHX). Various products, viz., ProRoot mixed with DW (PRW) or with CHX (PRC), Endocem mixed with DW (EW) or with CHX (EC), and Endocem premixed (EP) syringe type, were used. While antibacterial activity against *E. faecalis* was evaluated using a direct contact method, the specimens were stored in a shaking incubator for 30 d for antibacterial sustainability. The cytotoxicity was evaluated using a cell counting kit-8 assay in human periodontal ligament stem cells. The antibacterial activities of EP, EW, and EC were greater than those of PRC and PRW (*p* < 0.05). The antibacterial sustainability of EP was the highest without cytotoxicity for up to 30 days (*p* < 0.05). In conclusion, the pre-mixed injectable type EP was most effective in terms of antibacterial activity and sustained antibacterial effectiveness without cytotoxicity.

## 1. Introduction

Ever since the development of calcium silicate cement (CSC)—also known as Mineral Trioxide Aggregate (MTA)—by Dr. Torabinejad in the early 1990s, it has been widely used in various applications of endodontics, such as root canal obturation, perforation repair, and root-end filling [1,2]. Although materials such as zinc oxide eugenol, IRM, Cavit, and amalgam were used for root-end filling previously, these materials had drawbacks that included the potential for microleakage and associated bacterial growth [3]. CSC has become the gold standard for root-end filling not only owing to its biocompatibility but also because of its superior antibacterial effects compared to other materials [4,5].

*Enterococcus faecalis* (*E. faecalis*) is a Gram-positive facultative anaerobic bacterium that is one of the microorganisms most commonly associated with treatment failure during endodontic procedures [6]. *E. faecalis* is resistant to the high pH induced by calcium hydroxide [7]. Therefore, it is imperative that materials used in root-end fillings possess superior antimicrobial activities and sustained antimicrobial properties against *E. faecalis*. Previous studies have reported methods to enhance the antimicrobial properties of CSC [8,9]. One such approach is the incorporation of chlorhexidine (CHX), which is an antimicrobial agent that is widely used in dentistry, into CSC [10,11,12]. Although few studies have demonstrated the antimicrobial efficacy of CSCs mixed with CHX [11,13], generally there has been a lack of research on the sustained antimicrobial properties of CSCs, particularly against *E. faecalis*.

Even though CSCs are known to exhibit a high biocompatibility and sealing ability, they possess certain inherent limitations such as an extended setting time and difficulty in manipulation [14]. Recently, improved CSC types have been developed to overcome these drawbacks. Especially, ENDOCEM MTA (Maruchi, Wonju, Republic of Korea), which is a quick-setting type CSC with a setting time of 4 min as per the manufacturer’s specifications, offers several benefits owing to its rapid setting time. ENDOCEM premixed (Maruchi) was introduced as an alternative injectable-type CSC recently. This type of CSC offers several advantages, including the elimination of mixing requirements, convenient handling, and enhanced applicability. Various studies have been conducted on the physicochemical characteristics of these new CSC types [15,16,17,18]. Nevertheless, there is a notable dearth of studies that focus on evaluating the antibacterial activities of fast-set and premixed-type CSC. Especially, their sustained antibacterial effectiveness over 10 days has not been reported yet.

Therefore, this study aimed to compare the antibacterial activity and sustained antibacterial effectiveness for up to 30 days of several CSCs containing fast-set and premixed types. Additionally, the cytotoxicity of periodontal ligament stem cells and radiopacity were also evaluated. The null hypothesis is that there are no significant differences in antibacterial effect, sustained antibacterial effect, and radiopacity among the groups.

## 2. Materials and Methods

### 2.1. Materials and Preparation of the Specimens

The various types of MTA cement used in this study are listed in Table 1. One set of specimens was prepared by mixing ProRoot MTA with distilled water (PRW) and another set was prepared by mixing ProRoot MTA with chlorhexidine (PRC; Hexamedine, Bukwang Pharm. Co., Ltd., Ansan, Republic of Korea) at a water-to-powder (W/P) ratio of 0.18 cc/500 mg, according to the manufacturer’s instructions. Similarly, Endocem MTA was mixed with distilled water (EW) as well as chlorhexidine (EC) at a W/P ratio of 0.12 cc/300 mg. Because the injectable paste Endocem MTA Premixed Regular (EP) does not require mixing, it was injected directly into a silicone mold. Distilled water is used to mix the powder of CSC powders indicated in the instructions for use and was used in the PRW and EW groups as control groups in comparison with the use of CHX (PRC and EC groups). The specimens were prepared in the form of a disc with 6 mm diameter and 1 ± 0.1 mm thickness using a mold and allowed to set at 37 °C for 24 h in conditions of relative humidity above 90%. Subsequently, the specimens were sterilized by ultraviolet (UV) irradiation on a clean bench (JSCB-1200SB, JSR, Gongju, Republic of Korea) for 30 min. The distance from the UV light was approximately 50 mm.

### 2.2. Antibacterial Activity

The antibacterial activity of the specimens was evaluated against *E. faecalis* (KCTC) by adopting the direct contact method for the antibacterial test [19]. Bacteria were incubated aerobically in Brain Heart Infusion (BHI; BD DIFCO, Detroit, MI, USA) broth at 37 °C. After setting, each specimen was placed in the well of a 96-well plate (*n* = 6), and then each specimen was inoculated with 10 µL of *E.* faecalis (approximately 10^7^ bacteria); the specimens were then incubated at 37 °C for 1 h (Figure 1). Subsequently, 200 µL of BHI broth was added by pipetting and mixed, and 15 µL of the suspension was transferred to a new well plate filled with 215 µL of fresh BHI broth. After incubation at 37 °C for 6 h, the optical density (OD) values were measured to compare the bacterial growth of mid-log phase using a microplate reader (SpectraMax 250; Molecular Devices Co., San Jose, CA, USA). The OD values for each group were converted into percentages by considering the value of the negative control group as the base.

### 2.3. Sustained Antibacterial Effectiveness

To evaluate sustained antibacterial effectiveness, only those groups that exhibited antibacterial activity were chosen for testing. Disc shaped specimens similar to those used in the antibacterial assays were prepared (*n* = 6). Each specimen was placed in a 1.5 mL microtube filled with 1 mL PBS and stored in a shaking incubator at 120 rpm maintained at 37 °C for a predefined time duration of 1, 2, 10, 20, and 30 days. After expiry of the predefined time, the corresponding specimens were shifted to 96-well plates and subjected to the same process against *E. faecalis* using direct contact methods, as in the case of antibacterial assay. The OD values for each group were converted into percentages by considering the value of the negative control group as the base.

### 2.4. Cytotoxicity on Human Periodontal Ligament Stem Cell

Disc shaped specimens similar to those used in the antibacterial assays were prepared and eluted in 10 mL of the cell culture media (MEM Alpha, Gibco, Carlsbad, CA, USA) for 72 h at 37 °C according to ISO 10993-12:2012 [20]. Human periodontal ligament stem cells (hPDLSCs; Celprogen, Torrance, CA, USA) were distributed into 96-well plates at a density of 1 × 10^4^ cells/well and incubated for 24 h. The eluted media of each group was filtered through a 0.2 µm syringe filter (SC25P020SS, HYUNDAI MICRO Co., Ltd., Seoul, Republic of Korea) and applied to the cells using a method which is a slight modification of that suggested in ISO 10993-5 [21]. Briefly, hPDLSCs were seeded at a density of 1 × 10^4^ cells/well into a 96-well plate and incubated for 24 h. Subsequently, 100 µL of the extracted media was added to each well and incubated for 1, 2, and 3 days. Cell viability was measured using the Cell Counting Kit-8 (CCK-8; Dojindo Molecular Technologies, Rockville, MD, USA) following the manufacturer’s instructions. The OD value was measured at 450 nm using a microplate reader (SpectraMax 250). The negative control (NC) refers to cells that were cultured without any extract media.

### 2.5. Scanning Electron Microscopic Analysis

The surface morphology of all the specimens, i.e., those after the antibacterial assay as well as after sustained antibacterial effectiveness on the 10th day, were analyzed by field-emission scanning electron microscopy (FE-SEM; S-4800, Hitachi, Japan) at 5 kV under ×5000 and ×10,000 magnifications. Before observation, for fixation of the bacteria on the specimens, 2.5% glutaraldehyde (Sigma-Aldrich, St. Louis, MO, USA) was applied for 4 h. Then, the specimens were dehydrated by immersing them in 60, 70, 80, 90, and 100% ethanol (Sigma-Aldrich) for 10 min each. After drying the remaining ethanol, the specimens were coated using a platinum coater (E-1045; Hitachi Ltd., Tokyo, Japan).

### 2.6. Radiopacity

The radiopacities of the set materials were measured according to the test methods recommended in ISO 13116:2014 [22]. The specimens were disc shaped with 6 mm diameter and 1 ± 0.1 mm thickness (*n* = 6). To obtain radiographic images, each of the specimens were placed on a digital sensor along with an aluminum step wedge and exposed to an X-ray unit (Carestream Dental, Stuttgart, Germany) at 60 kV and 10 mA with a 300 mm focus-film distance. The gray pixel values of each specimen and the aluminum step wedge were determined using Image J (NIH, Bethesda, MD, USA). A linear equation was drawn using the grayscale of an aluminum stem wedge and the measured grayscale from the specimen was used to calculate the corresponding thickness of Al in millimeters.

### 2.7. Statistical Analysis

Statistical analyses were performed using IBM SPSS Statistics for Windows (version 26.0, IBM Corp., Armonk, NY, USA). Sustained antibacterial effectiveness data were analyzed by one-way analysis of variance (ANOVA) and Welch’s test with Games-Howell multiple comparisons (α = 0.05). Other data were analyzed using one-way ANOVA with Tukey’s multiple comparisons as a post hoc test (α = 0.05).

## 3. Results

### 3.1. Antibacterial Activity

The growth of *E. faecalis* was significantly inhibited in EW, EC, EP, and PRC (*p* < 0.05) (Figure 2). In particular, the EW, EC, and EP groups showed higher bacterial inhibition than the PRC group (*p* < 0.05) with no significant difference compared to the positive control (*p* > 0.05). Bacterial growth in PRW was not inhibited, thus indicating no antibacterial effects.

### 3.2. Sustained Antibacterial Effectiveness

Sustained antibacterial activity was evaluated only in the groups that exhibited antibacterial activity (Figure 3). The sustained antibacterial effectiveness of the EP and PRC groups was higher than that of the other experimental groups for up to 30 d (*p* < 0.05). More specifically, on day 20 the EP group showed an *E. faecalis* growth rate of less than 38% of the negative control group and more antibacterial effects than the PRC group (*p* < 0.05). Similarly, the growth rate of the EP group on day 30 was 52%, whereas that of the PRC group was 70%, although there were no statistically significant differences between the two groups (*p* > 0.05).

### 3.3. Cytotoxicity on Human Periodontal Ligament Stem Cell (hPDLSC)

Cytotoxicity was measured after 1, 2, and 3 days of incubation (Figure 4). There were no significant differences between the groups and the NC group (*p* > 0.05) after a day of incubation of the samples. On days 2 and 3, the PRC group exhibited the lowest cell viability among the groups (*p* < 0.05). The cell viability of the PRC group on day 3 was 66%, whereas that of the other groups were all over 70%.

### 3.4. Scanning Electron Microscopic Analysis

The scanning electron microscopic images of each specimen revealed various surfaces (Figure 5A). The PRC and EC particles were finer and exhibited a more uneven surface texture than those of PRW, EW, and EP. The number of *E. faecalis* attached to each specimen was different and fewer bacteria were observed in the specimens, as in EP (Figure 5B,C). This is consistent with the results of the antibacterial activity and sustained antibacterial effectiveness.

### 3.5. Radiopacity

Among all the specimens tested, the EP group showed the highest radiopacity (*p* < 0.05) (Figure 6). The radiopacities of ProRoot MTA and Endocem MTA were not affected by either of the mixing liquids, i.e., either distilled water or CHX (*p* > 0.05).

## 4. Discussion

Retrograde filling, also known as apicoectomy or root-end filling, is an endodontic treatment procedure that involves the removal of the apex of the tooth root and its replacement with biocompatible material [23]. When root canal treatment fails, the anatomical complex of the root-end region is removed and retrograde filling is performed to seal the apex and promote healing. An important characteristic of the materials used for retrograde filling is their ability to seal the root-end region and their biocompatibility [18]. Radiopacity of the apical region is also essential as it would allow an evaluation of the status of the applied material through radiography [24]. Ideally, the material should possess antibacterial activity and sustained antibacterial effects against bacteria such as *E. faecalis*, which often leads to the failure of root canal treatment [25]. Therefore, this study aimed to assess the antibacterial activity and sustainability of the antibacterial effect as well as the cell cytotoxicity and radiopacity of injectable premixed CSC and conventional CSCs when mixed with distilled water or CHX.

The antibacterial activities of the EW, EC, and EP groups were superior in comparison to those of the PRC. No antibacterial effects were observed in the PRW group. This may be attributable to differences in the characteristics of the constituents of CSCs [26,27]. CHX is an antiseptic solution used as a mouthwash or for topical treatment to prevent or treat periodontal disease and dental caries. Stowe et al. [28] showed that the antimicrobial properties of CSC were enhanced when mixed with CHX. Numerous other studies have also observed that the antibacterial activities of CSCs improve when mixed with CHX [12,28,29]. Therefore, the use of CHX with CSC in endodontic procedures can improve the sealing ability of CSC, reduce bacterial colonization, and promote faster healing of surrounding tissues [30]. In CSCs with resin, the addition of CHX showed no changes in tensile strength, cytotoxicity, water sorption, or solubility [31]. In this experiment, while CHX enhanced the antibacterial activity of ProRoot, we could not observe any change in antibacterial capability induced by CHX in Endocem as it already had antibacterial properties.

The cell viability of all the groups surpassed 70% of the negative group, with the exception of the PRC group on day 3 which was composed of the cells only. Despite the strong antibacterial effects of EP, EW, and EC, these groups did not exhibit cytotoxicity. A cell viability below 70% of the negative control is to be regarded as cytotoxic, according to the recommendations in ISO 10993-5 [21]. Other studies have established that CSC mixed with CHX increases apoptosis in gingival fibroblasts [10]. Previous studies on CSC primarily used dental pulp stem cells (DPSCs) because CSCs are often used for pulp capping [32,33]. However, in the context of retrograde filling, both DPSCs and PDLSCs play important roles [34]. Therefore, in the present study, we evaluated the cytotoxicity of PDLSCs. Although dental pulp stem cells and PDLSCs belong to the same mesenchymal stem cell lineage, they show differences in proteomics under osteogenic conditions [35]; they exhibit differences in the orientation of the sheet and stem cell markers as well [36]. CSCs used for root-end filling stimulate growth and proliferation of the PDLSCs, leading to improved healing and better tooth prognosis [37].

The sustained antibacterial effectiveness of EP was demonstrated for up to 30 days. Although few studies have investigated the sustained antibacterial effectiveness of CSCs [38,39], no reports have been published on their sustained antibacterial effectiveness beyond 10 days. Odabaş et al. [38] confirmed that the antibacterial activity against *E. faecalis* of CSC mixed with silver zeolite persisted for 72 h. According to Jafari et al. [39], MTA Filapex, a calcium silicate sealer, demonstrated antibacterial activity against *E. faecalis* that persisted for up to 7 days. Few bacteria attached to the surfaces of the EP and PRC specimens were also confirmed by the SEM images (Figure 5). It is worth noting that the EP group continued to exhibit sustained inhibition of *E. faecalis*: less than 38% on the 20th day and 52% on the 30th day.

Several methods to evaluate the antibacterial effects of CSCs have been reported. In a study by Kim et al. [40] that used a disc diffusion test, only EC showed an antibacterial effect on *E. faecalis* 2 h after setting [41]. In another disc diffusion test by Morita et al. [42], assessing the antibacterial activity after setting the CSC was difficult because of the white spots that were formed on the agar plate during the setting. There was a study that evaluated antibacterial activity by directly culturing with *E. faecalis* bacterial broth, however, this study could not identify any differences between the CSC groups [43]. Because the direct-contact test method is considered the most suitable for the comparison of antibacterial activities, it was adopted in this study [19].

The radiopacity of the EP group was the highest and the addition of CHX did not significantly change the radiopacity of ProRoot and Endocem MTA materials. As for the requirements of retrograde filling materials, they should display good radiopacity, possess low solubility, and exhibit appropriate physical properties, such as a low setting time and high compressive strength [27]. Moreover, the addition of CHX or other antibacterial agents should not deteriorate these properties. It is noteworthy that, until now, there are no significant studies related to the long-term physical and mechanical properties, and therefore, further studies are needed to confirm this hypothesis.

Endocem MTA was developed as a quick-setting alternative to address the shortcomings of conventional CSC [41]. Endocem MTA contains fine particles of pozzolan, a silicate-based material that reacts with calcium hydroxide that is formed during cement hydration [44]. As a slew of new products have been developed recently, various studies on their antibacterial activities, cytotoxicity, and physical and mechanical properties are in progress [18]. However, the physical and mechanical characteristics depending on the type of CSCs were not addressed in this study, which should be conducted in subsequent studies. To gain a deeper understanding of the bactericidal and bacteriostatic capabilities of CSCs, there is a subsequent need to conduct measurements using CFU or to perform LIVE/DEAD staining. This study is also limited to in vitro tests. In vivo studies would be required later for clinical applications since a variety of bacteria exist in a dynamic environment in oral conditions.

The antibacterial activity, sustained antibacterial effectiveness, and radiopacity of the fast-set type Endocem and the premixed type Encocem premixed were significantly different from those of the conventional CSC, ProRoot. Therefore, the null hypothesis was rejected. In this study, injectable EP showed excellent and sustained antibacterial activity without cytotoxicity, and the highest radiopacity.

## 5. Conclusions

In summary, CHX was effective in endowing antibacterial activity and sustained antibacterial effectiveness in the conventional CSC, ProRoot MTA that did not exhibit any antibacterial activity. The addition of CHX did not change the cell viability or radiopacity properties of ProRoot MTA and Endocem MTA, with the exception of cell viability on day 3 in the PRC group.

The pre-mixed injectable type, Endocem premixed, was the most effective CSC, considering its antibacterial activity, sustained antibacterial effectiveness, cytotoxicity, and radiopacity. Endocem premixed showed antibacterial activity and significant antibacterial effect against *E. faecalis* for 30 days with no cytotoxicity compared to other CSC groups. Therefore, it is suggested as a useful antibacterial material for root-end filling.

## Figures and Tables

**Figure 1 materials-16-06124-f001:**
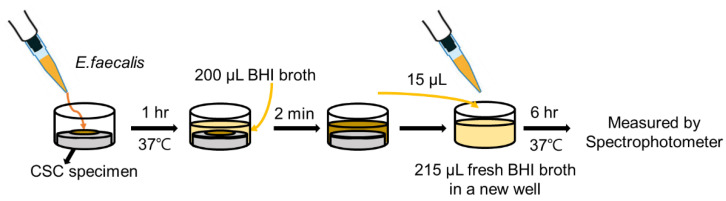
Schematic illustration of the antibacterial test.

**Figure 2 materials-16-06124-f002:**
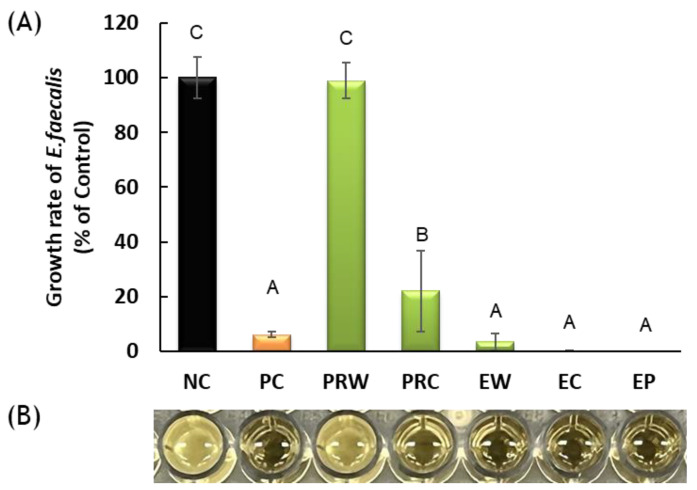
(**A**) Antibacterial activity of dental calcium silicate against *Enterococcus faecalis* by a direct contact test method. Different letters indicate significant differences between groups by one-way ANOVA with Tukey’s multiple comparisons (α = 0.05). (**B**) Photo of a row of 96-well plate after measuring the optical density. NC: *E. faecalis*, PC: Chlorhexidine. PRW: ProRoot with water, PRC: ProRoot with Chlorhexidine, EW: Endocem with water, EC: Endocem with Chlorhexidine, EP: Endocem Premixed.

**Figure 3 materials-16-06124-f003:**
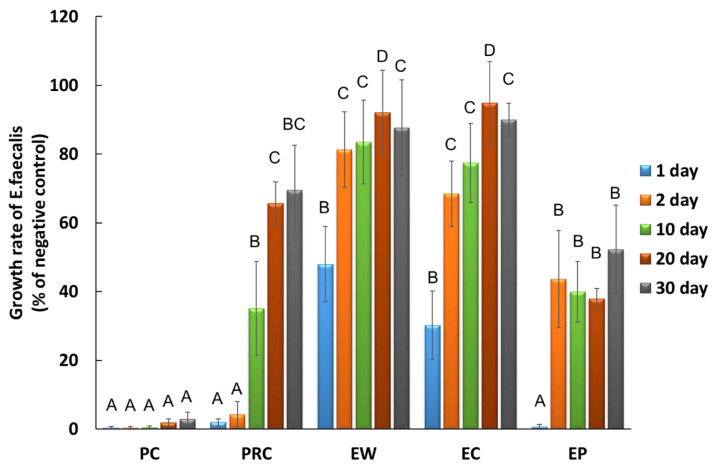
Sustained antibacterial effectiveness of calcium silicate cement against *Entertococcus faecalis*. Values are presented as the percentage of the negative control group that is composed of the bacterial only. Different letters indicate significant differences among the groups on the same day (among the same-colored bars) by one-way ANOVA and Welch with Games-Howell multiple comparisons (α = 0.05). Negative control: *E. faecalis* without calcium silicate cement. PC: Chlorhexidine, PRC: ProRoot with Chlorhexidine, EW: Endocem with water, EC: Endocem with Chlorhexidine, EP: Endocem Premixed.

**Figure 4 materials-16-06124-f004:**
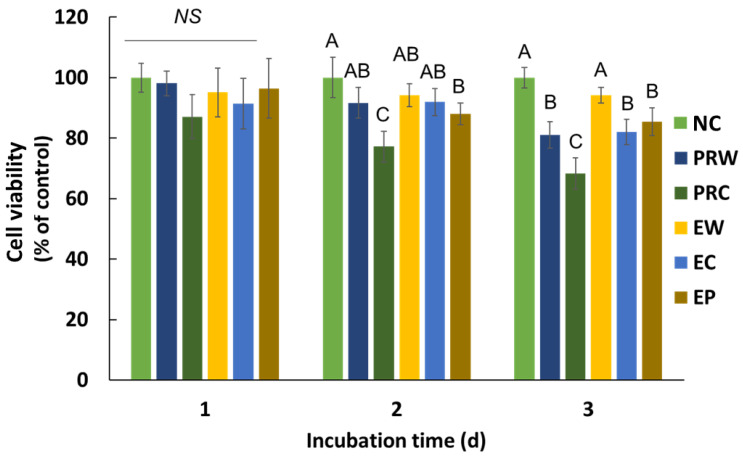
Cell viability on human periodontal ligament stem cells. Different letters indicate significant differences between groups on the same incubation day and NS means no significance by one-way ANOVA with Tukey’s multiple comparisons (α = 0.05). NC: cell only, PC: Chlorhexidine. PRW: ProRoot with water, PRC: ProRoot with Chlorhexidine, EW: Endocem with water, EC: Endocem with Chlorhexidine, EP: Endocem Premixed.

**Figure 5 materials-16-06124-f005:**
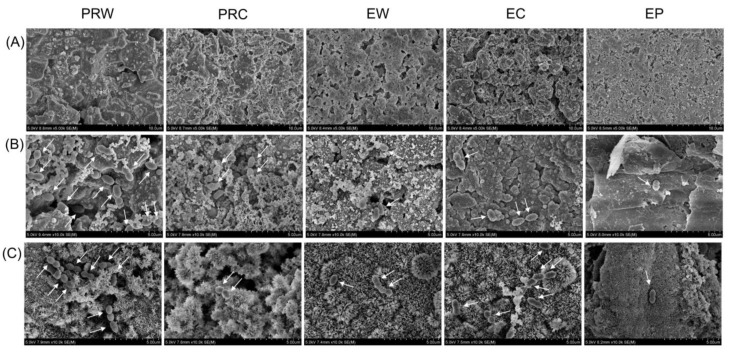
SEM images of (**A**) before the test, (**B**) after the antibacterial test, and (**C**) after the sustained antibacterial effectiveness test on day 10. PRW: ProRoot mixed with water, PRC: ProRoot with Chlorhexidine, EW: Endocem with water, EC: Endocem with Chlorhexidine, EP: Endocem Premixed. Arrows indicate *Enterococcus faecalis* adhesion. (**A**) 5000× magnification, (**B**,**C**) 10,000× magnification.

**Figure 6 materials-16-06124-f006:**
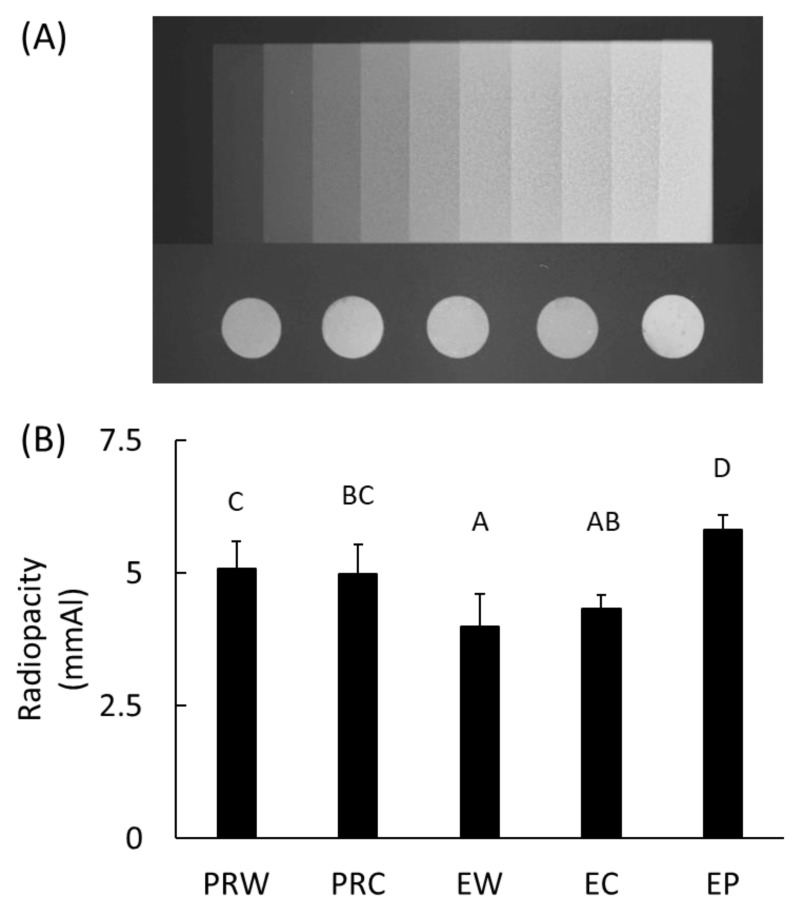
Radiopacity of the calcium silicate cements. (**A**) Representative photo of aluminum step wedge and specimens and (**B**) radiopacity of each group converted to the thickness of aluminum step wedge. Different letters indicate significant differences between groups by one-way ANOVA with Tukey’s multiple comparisons (α = 0.05). PRW: ProRoot with water, PRC: ProRoot with Chlorhexidine, EW: Endocem with water, EC: Endocem with Chlorhexidine, EP: Endocem Premixed.

**Table 1 materials-16-06124-t001:** Information on the calcium silicate cements used in this study.

Trade Name	Code	Liquid Type	Powder or Paste Composition	Manufacturer
Pro Root MTA	PRW	Distilled water	Tricalcium silicate, Dicalcium silicate, Bismuth oxide, Tricalcium aluminate, calcium sulfate dehydrate, tetra calcium aluminoferrite, gypsum, calcium oxide	Dentsply, Tulsa, TN, USA
	PRC	Chlorhexidine		
Endocem MTA	EW	Distilled water	Tricalcium silicate, Dicalcium silicate, Bismuth oxide, Tricalcium aluminate	Maruchi, Wonju, Republic of Korea
	EC	Chlorhexidine		
Endocem MTA Premixed Regular	EP	Injectable type paste	Zirconium dioxide, Calcium silicate, calcium aluminate, Calcium sulfate, Dimethyl sulfoxide, Lithium carbonate, Thickening agents	Maruchi, Wonju, Republic of Korea

## Data Availability

Data supporting the findings of this study are available from the corresponding author upon request.

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
