# Peer review of "Antibacterial Activity and Sustained Effectiveness of Calcium Silicate-Based Cement as a Root-End Filling Material against *Enterococcus faecalis"

_materials, 2023, doi:10.3390/ma16186124_

Round 1
Reviewer 1 Report
The authors must provide information on the initial concentration of cells.
Why did authors measure OD after 6h?
The method of OD shows only a change of medium optical density. You cannot get results on antibacterial activity based on the OD. CFU method is needed to confirm that. An alternative is also dead/live CLSM.
3.2 Sustained antibacterial effectiveness. What was negative control?
Figure 3: Results must be explained in more detail. The authors must explain how a material is antibacterial is there 70% growth after 30d?
Line 266 How sustained antibacterial effectiveness of EP and PRC can be stated if there is 45% and 70% bacterial growth after 30d?
Again, the authors cannot state that the material is antibacterial if only OD is measured. Therefore results, discussion and conclusions must be corrected.
Author Response
The authors must provide information on the initial concentration of cells.
à Thank you for comment, I added the number of bacteria.
L92: approximately 107 bacteria
Why did authors measure OD after 6h?
à Thank you for this comment, 6 hr is in the mid log phase of all group, therefore we can find difference among groups. And I explained it in the main text.
L:96 to compare the bacterial growth of mid-log phase
The method of OD shows only a change of medium optical density. You cannot get results on antibacterial activity based on the OD. CFU method is needed to confirm that. An alternative is also dead/live CLSM.
à We followed the procedures as described in the reference for our experiments. (Eldeniz, A.U.; Hadimli, H.H.; Ataoglu, H.; Orstavik, D. Antibacterial effect of selected root-end filling materials. J Endod 2006, 32, 345–349. DOI:10.1016/j.joen.2005.09.009.)
In this study, we did not elucidate the precise mechanism of whether a bacteriostatic effect, rather than a bactericidal one, appears over time. Future research, using methods such as CFU or LIVE/DEAD staining, is needed to clarify this point. This has been noted in discussion part 'limitation of the study' section
L319: To gain a deeper understanding of the bactericidal and bacteriostatic capabilities of CSCs, there is a subsequent need to conduct measurements using CFU or to perform LIVE/DEAD staining.
3.2 Sustained antibacterial effectiveness. What was negative control?
à Thank you for this comment, For sustained antibacterial effectiveness, negative control is the same as antibacterial test. I added that information.
L186: Negative control: E. faecalis without calcium silicate cement.
Figure 3: Results must be explained in more detail. The authors must explain how a material is antibacterial is there 70% growth after 30d?
Line 266 How sustained antibacterial effectiveness of EP and PRC can be stated if there is 45% and 70% bacterial growth after 30d?
à At 30 days, EP showed a statistically significant difference from the other groups, indicating an antibacterial effect by inhibiting bacterial growth. Naturally, as time progresses, antibacterial potency tends to weaken, so it cannot maintain the same efficacy as before.
For PRC, there was no statistical difference. Therefore, claiming antibacterial efficacy after 30 days for PRC may be a logical leap, so the content in the discussion has been revised."
L281 EP and PRC -à EP
Again, the authors cannot state that the material is antibacterial if only OD is measured. Therefore results, discussion and conclusions must be corrected.
à Thank you for your comment. Antibacterial ability is not only bactericidal, which kills bacteria, but also bacteriostatic, which inhibits bacterial growth. After 30 days, EP did not completely eliminate the bacteria, but it somewhat suppressed their growth, allowing for statistical differences between the groups. Hence, we termed 'antibacterial sustainability'.

Reviewer 2 Report
In the paper, the authors described an interesting research related to antibacterial activity and sustained effectiveness of calcium 2 silicate-based cement as a root-end filling material against 3 Enterococcus faecalis. However, what is missing are some explanations that the authors did not explain well enough and that must be done before publishing the paper:
1. The authors did not explain why they add water to improve the antibacterial properties of CSC? How can water have antibacterial properties and why did they even tested these samples?
2.The authors themselves state „Numerous other studies have also observed that the antibacterial activities of CSCs improve when mixed with CHX [12, 28, 29].“ So what is a novelty in this paper?
3. In general, the authors did not state what was new, what they did, and what sets this paper apart from others and makes it innovative enough to be published.
Author Response
The reply to the reviewer's comment is in the attached file.

Reviewer 3 Report
Title: good Abstract: - Please add the statistical test - please add a brief conclusions Introduction: - The originality is not provided and lots of studies were talked about the aims of the present study, please clarify - please add the null hypothesis and reject or accept it in the discussion Methods: - L 76: only temperature? are you sure? - which distance between the discs and the UV device? - Did the solubility of the discs or the released elements in the medium alter the measurement using the spectrophotometer? i think it is very difficult the measure the antibacterial activity of the bioceramic by using a spectro not a direct culture - L124-127: very poon, more info about the time ..... the discs were mixed??, i think the authors must used other words - Any sample size test? Results: - Figures: please always use small or capital letters for the stat - Figure 5: please add the magnification or scale bar - I think the authors have to place the Figure 5 after the text of SEM results and the same for Figure 6 - More info about the SEM observations Discussion: - Please clarify the limitations - Please use the following ref to discuss your results: Ashi, T.; Mancino, D.; Hardan, L.; Bourgi, R.; Zghal, J.; Macaluso, V.; Al-Ashkar, S.; Alkhouri, S.; Haikel, Y.; Kharouf, N. Physicochemical and Antibacterial Properties of Bioactive Retrograde Filling Materials. Bioengineering 2022, 9, 624. https://doi.org/10.3390/bioengineering9110624Author Response
The reply to the reviewer's comment is in the attached file.

Round 2
Reviewer 1 Report
The authors corrected all the issues.
Reviewer 2 Report
Authors do their best to answer the questions. The paper is generally well written, but regardless of the opinion that the results are lacking, I suggest that the paper be accepted for publication
Reviewer 3 Report
Good revision